# Fs Laser Patterning of Amorphous As_2_S_3_ Thin Films

**DOI:** 10.3390/ma17040798

**Published:** 2024-02-07

**Authors:** Claudia Mihai, Florin Jipa, Gabriel Socol, Adrian E. Kiss, Marian Zamfirescu, Alin Velea

**Affiliations:** 1National Institute of Materials Physics, Atomistilor 405A, 077125 Magurele, Romania; claudia.mihai@infim.ro; 2National Institute for Lasers, Plasma and Radiation Physics, Atomistilor 409, 077125 Magurele, Romania; florin.jipa@inflpr.ro (F.J.); gabriel.socol@inflpr.ro (G.S.); marian.zamfirescu@inflpr.ro (M.Z.); 3National Institute for Optoelectronics, INOE 2000, Atomistilor 409, 077125 Magurele, Romania; kadremil@yahoo.com

**Keywords:** femtosecond laser pulses, arsenic trisulfide (As_2_S_3_), photoexpansion and photoevaporation

## Abstract

This study investigates the morphological changes induced by femtosecond (fs) laser pulses in arsenic trisulfide (As_2_S_3_) thin films and gold–arsenic trisulfide (Au\As_2_S_3_) heterostructures, grown by pulsed laser deposition (PLD). By means of a direct laser writing experimental setup, the films were systematically irradiated at various laser power and irradiation times to observe their effects on surface modifications. AFM was employed for morphological and topological characterization. Our results reveal a clear transition threshold between photoexpansion and photoevaporation phenomena under different femtosecond laser power regimes, occurring between 1 and 1.5 mW, irrespective of exposure time. Notably, the presence of a gold layer in the heterostructure minimally influenced this threshold. A maximum photoexpansion of 5.2% was obtained in As_2_S_3_ films, while the Au\As_2_S_3_ heterostructure exhibited a peak photoexpansion of 0.8%. The study also includes a comparative analysis of continuous-wave (cw) laser irradiation, confirming the efficiency of fs laser pulses in inducing photoexpansion effects.

## 1. Introduction

Amorphous chalcogenide materials consist of a chalcogen element from group 6A of the periodic table, such as S, Se, or Te, covalently bonded to one or more network former elements from groups 3–5A, such as As, Sb, Ge, Si, Ga, etc. [1]. In recent years, chalcogenide materials have gained increasing interest due to their high refractive index [2], wide transparent window [3], ultrahigh optical nonlinearity [4], ultrafast response time to light, and photosensitivity [5]. These outstanding properties are making chalcogenides promising candidates for diverse applications, including photonic devices [6], polarization-sensitive optical elements [7], infrared optical imaging [8], fast-expanding metamaterials [9], as well as non-volatile memory and switching devices [10,11]. In addition to bulk chalcogenide glasses, thin films and nanoscale waveguides [12] have been extensively explored lately.

Femtosecond (fs) lasers can modify the physical and chemical properties of transparent glasses and amorphous films by direct writing due to nonlinear absorption phenomena [13,14,15]. The permanent induced modifications by focusing the laser on the surface of the materials, such as photoexpansion [16], crystallization [17], or chemical composition changes caused by ion migration [18], can be used to build periodic structures and waveguides. The photoexpansion is influenced by the laser fluence and is accompanied by a change in refractive index [19] and an enhanced optical nonlinearity [20]. The probability of obtaining a photoexpansion effect using IR photons depends on the radiation intensity. IR lasers that work in a continuous wave (cw) regime cannot achieve high photon density without inducing thermal damage in the material. For this reason, low-energy short-pulses IR lasers are focused on the surface of the material. CW IR lasers often encounter difficulties in achieving increased photon densities without inducing thermal damage. It is important to note that the pulse width influences these interactions. Ultra-short pulse lasers, due to their transient duration, facilitate obtaining high photon densities while concurrently mitigating thermal effects. This distinction is essential as the photon density fundamentally determines the type of ionization mechanism employed, encompassing multiphoton, avalanche, or tunneling ionization processes. When a femtosecond laser pulse is focused on the surface, high power density can be reached. The focused laser beam usually exhibits a Gaussian intensity distribution where the intensity is highest at the center and decreases towards the edges, influencing the interaction dynamics with the material. The photoexpansion effect is, therefore, induced in a very small volume that can be smaller than the size of the focused spot.

Chalcogenide glasses are known for their broad transmission window in infrared [21], which makes them appropriate for photonic applications [22]. Arsenic trisulphide (As_2_S_3_) is intensively studied due to its optical properties and versatility in structural modifications [23,24]. This material is characterized by a high nonlinear refractive index, high transmission in infrared regions, and low phonon energies. Amorphous As_2_S_3_ has a transparency window that ranges between 0.7 μm and 10 μm [25], a bandgap of 2.4 eV [26], and a refractive index of 2.5 [27]. When irradiated with bandgap or intense sub-bandgap light, the surface morphology of As_2_S_3_ thin films changes due to photoexpansion [28], photoevaporation/photodepression [29], or ablation [30] processes. Moreover, the presence of gold in gold\amorphous chalcogenide heterostructures can influence the optical and structural properties, but also the volume changes of the material [31]. Metal photodissolution in chalcogenide films facilitates achieving dry grayscale lithography [32,33,34], offering a new powerful class of photoresists for versatile lithography since a greater etch selectivity than in organic photoresists was demonstrated. The addition of a gold layer to As_2_S_3_ films could enable surface plasmon resonance phenomena [35], which could be exploited in a variety of applications, including sensing and imaging.

In this study, we investigate the patterning of periodic structures in As_2_S_3_ amorphous thin films and gold\amorphous As_2_S_3_ heterostructures by laser irradiation. After the laser irradiation with different laser powers and for different exposure times in both continuous and femtosecond pulsed modes, the film morphological evolution was extensively characterized using atomic force microscopy. The shape of the modified surface is found to be a function of the laser power, while the presence of gold influences the threshold between photoexpansion and photovaporization. This study provides a comprehensive analysis of how laser parameters and material composition interact to influence photoexpansion and other morphological changes.

## 2. Materials and Methods

Thin amorphous arsenic trisulfide films were grown by pulsed laser deposition (PLD) on a glass substrate using a KrF* laser source (COMPexPro 205, Lambda Physics-Coherent, Santa Clara, CA, USA) with a wavelength λ = 248 nm and τ_FWHM_ = 25 ns. The bulk As_2_S_3_ target was irradiated using a laser fluence of 1.5 J/cm^2^, and the repetition rate was 10 Hz. The depositions were carried out in a PLD chamber at room temperature in a vacuum, with the residual gas pressure of 3 × 10^−6^ Torr. Ten thousand laser pulses were applied onto the target positioned parallel to the substrate at a distance of 4 cm in order to obtain the films with a thickness of 1 µm. Moreover, glass substrate\Au\As_2_S_3_ heterostructures were also prepared, where the Au layer is 100 nm in thickness.

The As_2_S_3_ films and Au\As_2_S_3_ were processed using a standard Direct Laser Writing (DLW) setup consisting of a Ti:Sapphire (Synergy Pro, Newport Corporation, Santa Clara, CA, USA) femtosecond laser oscillator with 800 nm central wavelength, 5 nJ pulse energy, 15 fs pulse duration, and 80 MHz repetition rate. The micro-processing setup is also equipped with a dispersion compensation module and a spatial filter needed to preserve the pulse duration and the spatial profile. A 100× IR Mitutoyo microscope objective with a numerical aperture (NA) of 0.5 was used for microfabrication of the structures. The laser beam has a Gaussian spatial profile with an estimated diameter spot in the focal plane of 3 µm. However, the minimum size of the features that can be produced on the film by laser irradiation is influenced by the laser power and the exposure time. The laser power is adjusted using a variable attenuator consisting of a half wave-plate and two reflection polarizers. The laser beam is fixed, while the film is shifted in XY directions using a piezoelectric translation stage with nanometer positioning resolution controlled by the software according to the designed scanning path and speed. The laser focus on the sample surface was monitored using a CCD camera.

Several processing schemes were used. A network of periodical points was created by local exposure of individual spots separated by 5 µm × 5 µm. The irradiation was conducted for exposure ranging from 50 ms to 400 ms at each point, corresponding to approximately 4 × 10^6^ pulses per spot for 50 ms and 32 × 10^6^ pulses per spot for 400 ms, where the laser pulse duration was 15 fs. The laser power was varied from 0.125 to 2 mW. At a power setting of 1 mW, the laser intensity would be approximately 11.8 MW/cm^2^. The same IR laser was used in a continuous wave (cw) regime. Lines using a scanning speed between 50 and 100 µm/s and a cw irradiation were also inscribed.

Atomic force microscopy (AFM) investigations were conducted with an Innova (Bruker, Billerica, MA, USA) instrument. In order to obtain high-quality images, the microscope was vibration-damped. Phosphorus (n) doped silicon pyramidal tips (Veeco, RTESPA model) mounted on a 125 µm rectangular cantilever were used. The measurements were carried out in tapping mode, in a forward direction, with a scan speed ranging from 2.5 µm/s (0.5 Hz) to 8 µm/s (0.4 Hz), depending on the image size.

## 3. Results and Discussion

Following the IR laser irradiation protocol described above, both the As_2_S_3_ thin films and Au\As_2_S_3_ heterostructures were processed. The surface of non-irradiated samples is uniform and smooth without any morphological details. After exposure to intense laser pulses in the femtosecond regime, the morphology of the irradiated zones changed as a function of laser power and irradiation time. The AFM images of the fs laser-irradiated samples are given in Figure 1.

For lower laser powers (below 0.5 mW) and shorter irradiation times (below 400 ms), the surface of the As_2_S_3_ films was not altered. Changes on the As_2_S_3_ thin film surface were induced at 0.125 mW and 400 ms irradiation time, respectively. Figure 1 shows only the microstructures obtained between 0.5 and 2 mW for comparison purposes. Several features can be observed, with the most interesting phenomena at the transition between photoexpansion and photoevaporation. A clear transition threshold from photoexpansion to photoevaporation is observed between 1 and 1.5 mW independent of the exposure time; however, for 1 mW and 400 ms, it seems that both effects are present (i.e., the microstructure is half hole, half hillock). The shape of the microstructure also changed before the transition from higher and thinner to shorter and larger configurations. After transition, the hole size enlarges proportionally to the increase of the laser power, and evaporated material starts to be visible on the surface of the film. It is worth noting that at 1 mW laser power and 100 ms exposure time, a double structure can be observed, while at 1.5 mW and 200 ms, a small fragment of the hillock remains in the volcano-like hole. These are indications of how the transition from photoexpansion to photoevaporation takes place. It seems to be an asymmetric phenomenon, and it depends on the irradiation parameters. The asymmetry could be related to the photoinduced optical anisotropy observed in chalcogenide glasses [36,37].

In the case of As_2_S_3_ bulk glass, the increase of the volume under laser irradiation exposure is due to photostructural changes and can be as high as 2%. On the other hand, when the irradiated volume is constrained by the femtosecond pulses, different factors, such as the stress at the interface, lead to a higher photoexpansion effect. Giant photoexpansion of more than 4% was observed in amorphous As_2_S_3_ thin films irradiated with bandgap and sub-bandgap light [28]. The morphological changes are, in some cases, more prominent for sub-bandgap irradiation because they are induced by light with photon energies in Urbach-edge regions or in the weak absorption tails that extend below the Urbach edge as in As_2_S_3_ [38]. The weak absorption tails are caused by the unoccupied gap states below the conduction band, which are produced by the homopolar As–As wrong bonds [39]. Two-photon absorption is responsible for photomodification when relatively low-energy laser pulses, as compared with conventional single-photon absorption in cw laser writing, are used [40].

At higher laser power, photoevaporation occurs by the elimination of clusters of different sizes and compositions. Usually, photoevaporation is preceded by photoinduced fluidity caused by an athermal photoelectronic excitation process, which produces the transformation of atomic bonds [41].

A detailed analysis of the height and depth profile of the irradiated spots is given in Figure 2. The height of the structures is 52 nm for 0.5 mW and 50 ms laser parameters and increases as the irradiation time increases up to a maximum of 100 nm for 400 ms. A different behavior is obtained for 1 mW; a decrease in height can be observed from 112 nm to 66 nm with the increase in exposure time. A double peak curve can be observed for 400 ms, which evidences a transition from photoexpansion to photoevaporation. A double peak hole is then observable for 1.5 mW and exposure times up to 200 ms, ending with a volcano-shaped hole for 400 ms with a cone as high as 41 nm and a hole as deep as 49 nm. For 2 mW, the height of the cone decreases, and the depth of the hole deepens with the increase of the exposure time, reaching 133 nm for 100 ms. In the case of larger exposure times (i.e., 200 and 400 ms), the surface of the film is covered by debris from the material, and it becomes increasingly difficult to perform accurate measurements.

Each line profile, for both hillocks and holes, was fitted using a Gaussian function
$$(y=y_{0}+\frac{Ae^{\frac{-4ln(2){\left(x-x_{c}\right)}^{2}}{w^{2}}}}{w\sqrt{\frac{\pi }{4ln(2)}}}),$$ where *A* is the area of the hillock/hole without a double peak while *w* is the full-width half maximum (FWHM). The obtained results are summarized in Table 1. Additionally, the volume photoexpansion for the hillock s was computed using the expression:$$\varepsilon_{V}=100\frac{V}{\pi r^{2}d}$$
where *V* is the volume of the hillock and $\pi r^{2}d$ is the volume of a cylinder below the hillock with the same radius *r* [42]. *d* is the thickness of the As_2_S_3_ thin film. The volume of the hillock is estimated using the relation:$$V=\pi \int_{0}^{r}{f(x)}^{2}dx$$
where *f(x)* is the Gaussian function.

The order of magnitude of photoexpansion corresponds to the size of the hills observed on the surface of the laser-irradiated area of the As_2_S_3_ thin films, with a maximum value of 5.2% obtained for 1 mW and 50 ms or 0.5 mW and 100 ms. The observed maximum value of 5.2% for both 1 mW at 50 ms and 0.5 mW at 100 ms settings suggests that the cumulative power density, or dose, is equivalent for these configurations. This equivalence is derived from the product of the number of pulses per spot and the intensity. The explanation of the photoexpansion process could be related to the charge redistribution on the sulfur atoms of each cluster induced by the femtosecond laser pulses. The charge redistribution increases the electrical repulsion between sulfur atoms, leading to an expansion of the clusters by network reconfiguration without breaking any bonds [43]. These hillocks act as microlenses for the red and near-infrared spectrum of electromagnetic radiation and can be used in two-dimensional optoelectronic circuits [44]. The maximum photoexpansion is comparable with the one observed in Ga_2_S_3_ films (i.e., 2.45%) [42] but is obtained for a much lower laser power of only 1 mW compared to 100 mW for Ga_2_S_3_. The explanation could be the fact that As_2_S_3_ films are suggested to consist of hard sphere As4S6 molecules that, when heated or illuminated, polymerize or cross-link to form a network structure characteristic of the bulk glass. This could potentially make As_2_S_3_ more responsive to lower laser powers [40]. Moreover, As_2_S_3_ has been observed to have significant differences between the dynamics of metastable/permanent and transitory effects. The permanent photoexpansion is much larger in magnitude and slower in kinetics than the transient or reversible changes. This could imply that As_2_S_3_ has inherent properties that make it more susceptible to photoexpansion at lower laser powers [45].

The results obtained by cw laser irradiation of the As_2_S_3_ films are shown in Figure 3. These unveil that the photoexpansion effect without any traces of photoevaporation occurs at laser power values up to 2 mW without any traces of photoevaporation.

A comparative analysis of the height of the irradiated areas for different irradiation speeds is given in Figure 4. We applied a similar analysis to the cw laser irradiation, and the parameters are given in Table 2. The height of the lines is the highest, 51 nm, for 50 µm/s at 1.5 W, leading to a maximum photoexpansion of 2.4%. This is proof that the fs pulses are more efficient to maximize the photoexpansion. However, some similarities with the fs laser regimes are retained. It can be observed that as the irradiation speed increases, the maximum height of photoexpanded volume moves to lower laser power (i.e., from 1.5 mW at 50 µm/s to 1 mW at 100 µm/s). Moreover, the width of the lines is the largest for the highest laser power (i.e., 2 mW). At slower speeds and higher power, the material might reach a saturation point beyond which additional energy does not contribute to further photoexpansion. When the speed is increased, the material may not reach this saturation point, allowing for more efficient use of the laser power. Additionally, the interaction between light and material is highly nonlinear in As_2_S_3_. The relationship between irradiation speed, power, and resulting morphology could be influenced by a variety of nonlinear optical effects. An asymmetric profile is observed for 100 µm/s and 2 mW, with a shoulder that resembles the double peak shape obtained for pulsed radiation. The maximum photoexpansion obtained by the fs laser is two times higher than the photoexpansion obtained with the cw laser. While Tanaka et al. [46] suggest that a high repetition rate laser (such as 80 MHz) can be regarded as quasi-continuum irradiation, it is important to distinguish this from a truly continuous process. With an 80 MHz repetition rate and 15 fs pulse widths, there remains a significant factor of 8 × 10^4^ between the temporal periodicity of the laser pulses and the interaction time itself. This difference highly influences the laser-matter interaction dynamics, favoring non-linear mechanisms in the ultra-short process as opposed to the primarily heat diffusion-driven interactions in a continuous process.

In order to study the influence of a thin gold interlayer, we reproduced the laser irradiation experiments on Au\As_2_S_3_ heterostructures. The morphological modifications of Au\As_2_S_3_ multilayers irradiated with fs laser radiation are given in Figure 5. The photoexpansion phenomenon is induced at a lower laser power value (i.e., 0.125 mW) and for a shorter irradiation time (i.e., 100 ms), while photoevaporation is present already at 1 mW.

The images are noisy due to the roughness of the gold film of 1.5 nm. A summary of the results on the height of the irradiated spots is given in Figure 6 and Table 3. The height of the structures is 10 nm for 0.125 mW and rises as the irradiation power increases up to a maximum of 20 nm for 0.5 mW and 100 ms (which means a photoexpansion of 0.79%). The photoexpansion is 3.3 times lower than the single As_2_S_3_ films at the same irradiation parameters (i.e., 100 or 400 ms). The fs laser power threshold for the photoevaporation is 1 mW. At longer exposure times of 400 ms, the height of the structures remains relatively stable across different laser powers, ranging from 10 nm to 16 nm. This suggests that the presence of the gold layer influences the threshold for photoexpansion and photoevaporation, possibly due to enhanced energy absorption.

The data indicates that the gold layer seems to lower the threshold for photoexpansion, as evidenced by the onset of changes at lower laser powers and shorter exposure times compared to the As_2_S_3_ films without the gold layer. However, the gold layer also appears to limit the extent of photoexpansion at higher laser powers, as seen in the reduced heights of the hillocks and photoexpansion values. This could be due to a variety of factors, such as altered optical properties or changes in the stress–strain relationship due to the gold layer. The gold layer might be affecting the electronic structure of the As_2_S_3_ film, leading to different absorption characteristics and subsequent material responses. On the other hand, the gold layer could also serve as a mechanical support that restricts the deformation of the As_2_S_3_ film, thereby affecting its stress–strain relationship. This could be particularly relevant because the gold layer is more rigid than the As_2_S_3_ layer, as it could limit the extent to which the As_2_S_3_ can expand.

Finally, the outcomes of the cw laser irradiation for the Au\As_2_S_3_ films are depicted in Figure 7. It can be observed that the photoexpansion effect without any traces of photoevaporation occurs only at fs laser power values up to 1.5 mW, without any traces of photoevaporation.

The results of the morphological analysis on the irradiated dots are given in Figure 8 and Table 4. A different behavior was observed under cw laser exposure in the case of Au\As_2_S_3_ compared with simple As_2_S_3_ films. At a laser power of 0.5 mW and an irradiation speed of 50 µm/s, the height of the structures is 7 nm with a photoexpansion of 0.3%. Moreover, when increasing the laser power to 1 mW and at the same speed, the height was not significantly altered, but the photoexpansion slightly increased to 0.3%. However, a notable change is observed at 1.5 mW and 50 µm/s, where the height rose to 18 nm with a photoexpansion of 0.5%.

When the irradiation speed is doubled to 100 µm/s, the height of the structures at 0.5 mW increases to 16 nm, with a photoexpansion of 0.7%. At 1 mW, the height slightly decreases to 14 nm, but the photoexpansion remains almost constant at 0.6%. Further increasing the laser power to 1.5 mW at this speed results in a decrease in both height and photoexpansion, registering 11 nm and 0.5%, respectively.

These results indicate that the fs pulses are more efficient for maximizing photoexpansion in As_2_S_3_ films, but the presence of the gold layer in Au\As_2_S_3_ seems to alter this efficiency. The gold layer appears to introduce complexities in the stress–strain relationship and energy absorption characteristics of the material.

The results for both fs laser pulses and cw laser irradiation are summarized in Figure 9. The onset of photoexpansion in the Au\As_2_S_3_ heterostructures occurred earlier, at 0.125 mW and 100 ms, compared to the As_2_S_3_ films, where it induced at 0.125 mW and 400 ms. This suggests that the gold layer may be playing a role in lowering the threshold for photoexpansion. The As_2_S_3_ films exhibited a maximum photoexpansion of 5.2% at 1 mW and 50 ms, while the Au\As_2_S_3_ heterostructures showed a maximum of only 0.8% obtained at lower power (0.25–0.5 mW) and longer irradiation time (100 ms). The presence of a gold layer beneath the As_2_S_3_ layer can affect the photoexpansion behavior due to the Au’s ability to enhance the electromagnetic field at the interface through localized surface plasmon resonance [34]. This enhancement can increase the efficiency of light absorption in the As_2_S_3_ layer, leading to the observed effects at lower power and shorter irradiation times. Regarding the influence of temperature increase due to laser heating on the photoexpansion effect, researchers have computed the temperature rise and thermoelastic expansion in As_2_S_3_ glass, indicating that the contribution of thermal expansion cannot be excluded [47]. However, this model primarily applies to large-dimension samples under the semi-infinite sample approximation. For longer illumination and finite-sized samples, interactions with the surrounding medium and elastic boundary conditions significantly impact thermal expansion. Numerical simulations revealed that the amplitude of thermoelastic deformation is negligible as compared to photoexpansion, suggesting a predominantly non-thermal origin [45]. Additionally, phenomena like photoinduced fluidity in chalcogenide glasses manifest athermally at low temperatures [41,48], further supporting non-thermal mechanisms. Finally, femtosecond lasers minimize thermal effects due to their inherent nonlinear light–matter interaction [23].

Both types of samples exhibited some form of asymmetry in their morphological features, but it was more pronounced in the As_2_S_3_ sample. This could be related to the photoinduced optical anisotropy observed in chalcogenide glasses. Regarding the cw laser irradiation, it can be observed that for As_2_S_3_ film, the photoexpansion is the highest for a lower irradiation speed. At slower speeds, the material has more time to dissipate heat between laser pulses. When the speed increases, the material may not have enough time to cool down, leading to a cumulative thermal effect. This could mean that less power is needed to achieve similar or greater effects. On the other hand, for the Au\As_2_S_3_ heterostructure, this is not the case, confirming again the hypothesis of the Au layer as an efficient heat sink. These findings have significant implications for the development of photonic devices and/or two-dimensional optoelectronic circuits by femtosecond laser lithography.

## 4. Conclusions

In this study, we have systematically investigated the effects of fs and cw laser irradiation regimes on As_2_S_3_ thin films and Au\As_2_S_3_ heterostructures. The As_2_S_3_ films exhibited a clear transition threshold from photoexpansion to photoevaporation between 1 and 1.5 mW when fs laser pulses were employed, independent of exposure time. Specifically, a maximum photoexpansion of 5.2% was achieved at a laser power of 1 mW. In contrast, the Au\As_2_S_3_ samples showed more complex behavior, with the gold layer appearing to modulate both the stress–strain relationship and energy absorption characteristics. The maximum photoexpansion observed in the Au\As_2_S_3_ heterostructure was 0.8%. Notably, fs pulses at a power of 1 mW and 400 ms were over four times more efficient in maximizing photoexpansion in As_2_S_3_ films compared to the Au\As_2_S_3_ heterostructure. This efficiency was altered in the presence of the gold layer, requiring higher laser powers to achieve similar effects. These findings suggest that the laser–material interaction mechanisms involved during the photoexpansion generation in As_2_S_3_ films are also influenced by the nature of the substrate, potentially offering both challenges and opportunities for optoelectronic applications.

## Figures and Tables

**Figure 1 materials-17-00798-f001:**
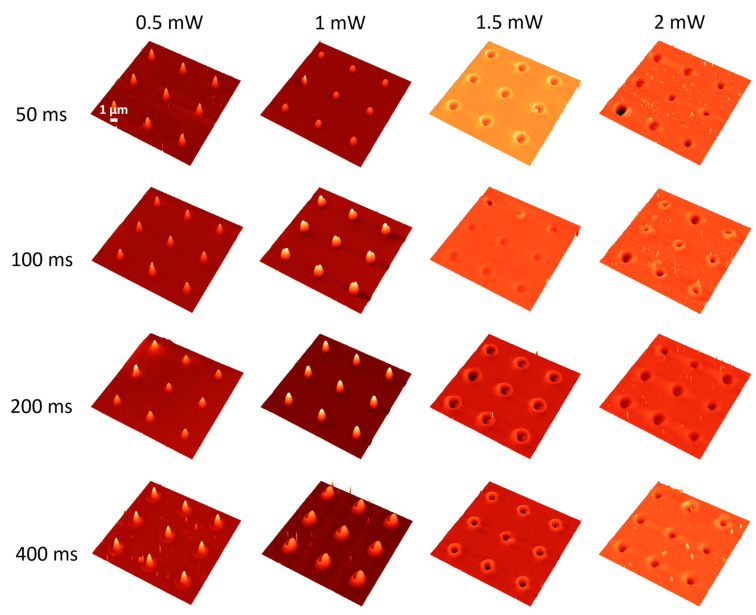
AFM images of the most relevant laser-irradiated areas of the As_2_S_3_ film with femtosecond laser pulses. The laser power is given for each column on top of the image, while the exposure time is shown on the left-hand side.

**Figure 2 materials-17-00798-f002:**
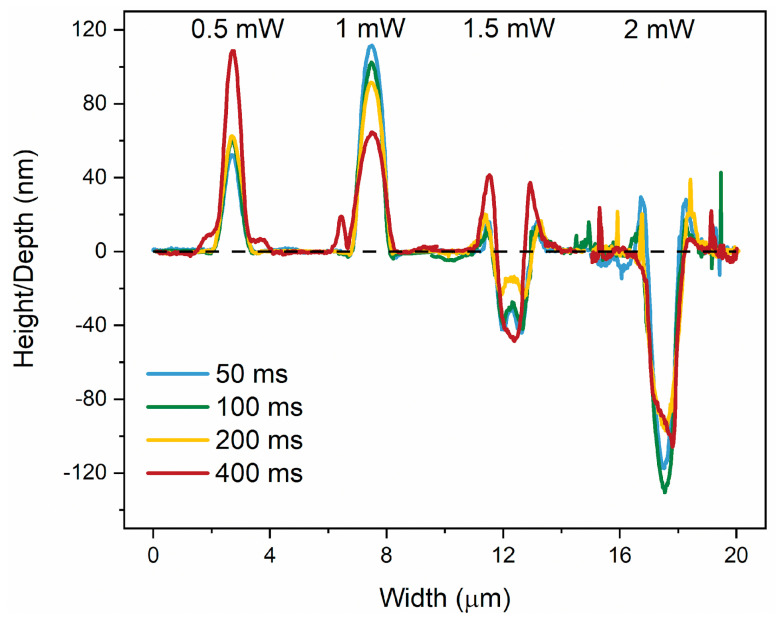
Line profiles of the most representative irradiated areas of the As_2_S_3_ film with femtosecond laser pulses for each laser power and irradiation time.

**Figure 3 materials-17-00798-f003:**
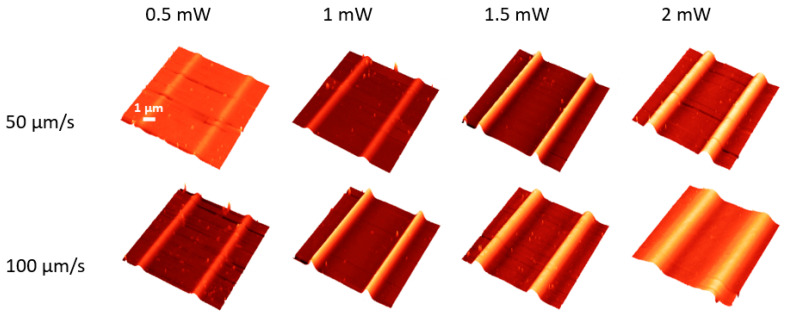
AFM images of the most relevant laser-irradiated As_2_S_3_ films with continuous wave laser. The laser power is given for each column on top of the image, while the scan speed during the irradiation speed is shown on the left-hand side.

**Figure 4 materials-17-00798-f004:**
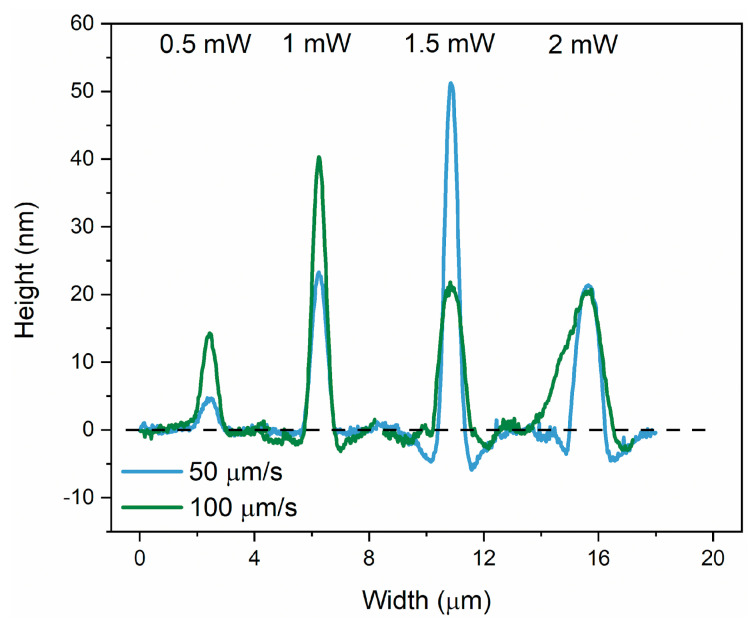
Line profiles of the representative irradiated areas of the As_2_S_3_ film at each cw laser power and irradiation speed.

**Figure 5 materials-17-00798-f005:**
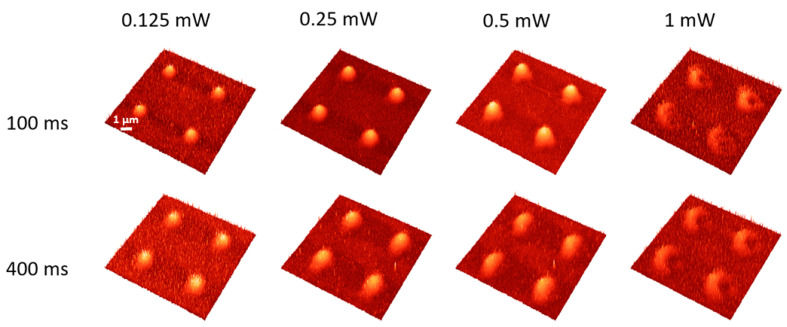
AFM images of the most relevant fs laser-irradiated areas of the Au\As_2_S_3_ heterostructures. The laser power is given for each column on top of the image, while the exposure time is shown on the left-hand side.

**Figure 6 materials-17-00798-f006:**
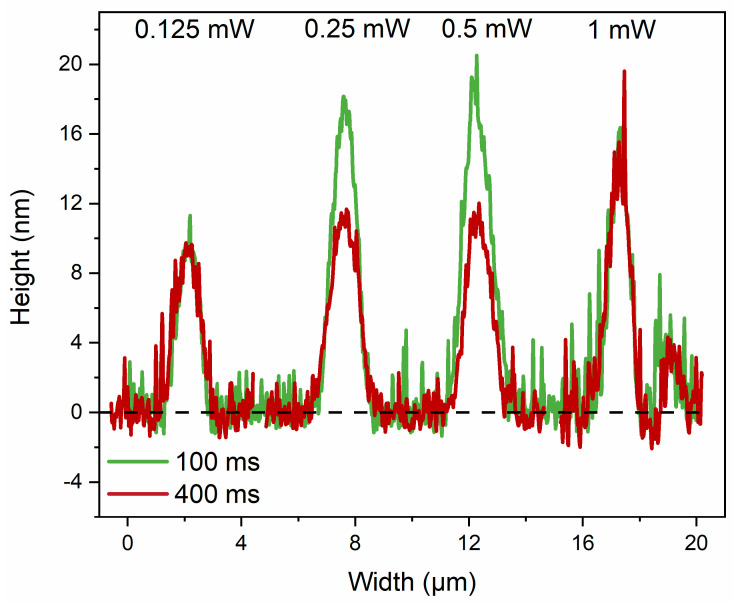
Line profiles of the most representative irradiated areas of the Au\As_2_S_3_ heterostructures with fs laser pulses for each laser power and irradiation speed.

**Figure 7 materials-17-00798-f007:**
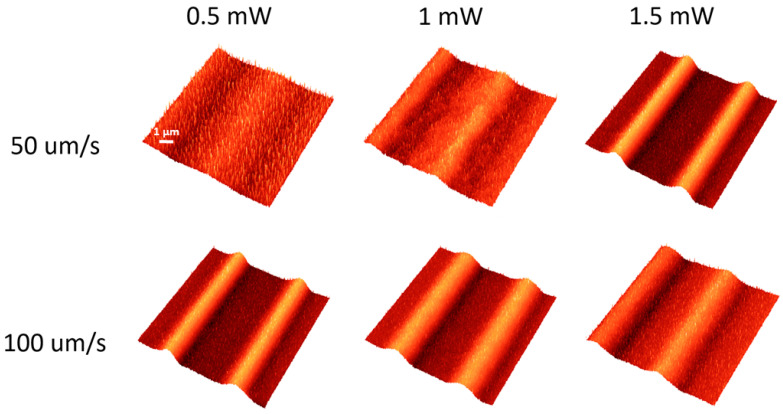
AFM images of the most relevant laser-irradiated Au\As_2_S_3_ multilayers with cw laser. The laser power is given for each column on top of the image, while the irradiation speed is shown on the left-hand side.

**Figure 8 materials-17-00798-f008:**
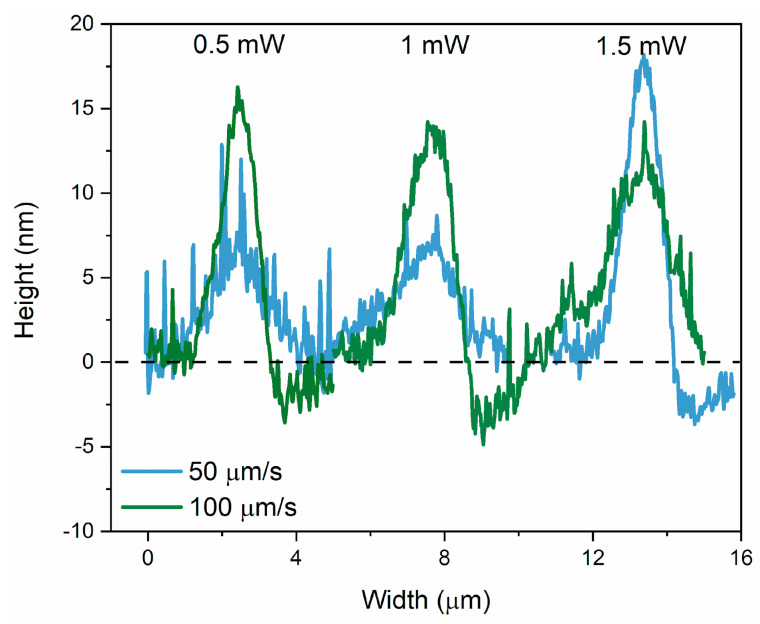
Line profiles of the cw laser photoinduced structure on the Au\As_2_S_3_ heterostructures for the most representative laser power and irradiation speed values.

**Figure 9 materials-17-00798-f009:**
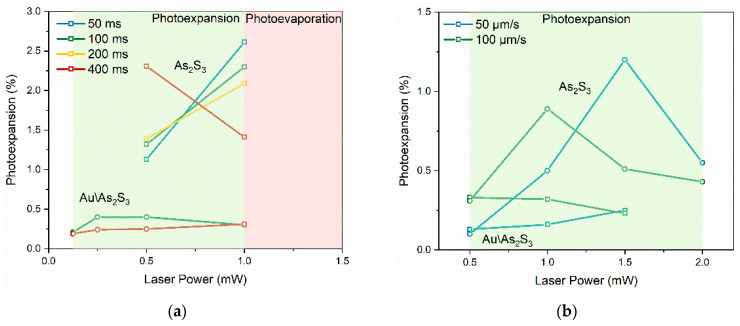
Summary of the laser-induced modifications as a function of energy and duration for (**a**) fs laser pulses and (**b**) cw laser irradiation in both As_2_S_3_ and Au\As_2_S_3_ samples.

**Table 1 materials-17-00798-t001:** Fitting parameters of each hillock/hole of the As_2_S_3_ film irradiated with fs laser pulses using a Gaussian function. The volume photoexpansion ($\varepsilon_{V}$) and height/depth (*h*/*d*) of the irradiated microstructures is also given. The calculated photoexpansion refers to an arsenic trisulfide film with a thickness of 1 µm.

Average Laser Power (mW)		0.5	1	1.5	2
Laser Fluence (µJ/cm^2^)		88.5	176.9	265.4	353.9
Laser Intensity (MW/cm^2^)		5.9	11.8	17.7	23.6
Exposure Time	Computed Parameter				
50 ms	*A* (µm^2^)	0.034	0.097	0.043	0.092
	*w* (µm)	0.51	0.63	0.72	0.56
	$\varepsilon_{V}$ (%)	2.3	5.2	-	-
	*h/d* (nm)	52	112	44	119
100 ms	*A* (µm^2^)	0.039	0.084	0.038	0.111
	*w* (µm)	0.50	0.62	0.73	0.63
	$\varepsilon_{V}$ (%)	5.2	4.6	-	-
	*h/d* (nm)	62	103	46	133
200 ms	*A* (µm^2^)	0.046	0.080	0.026	0.097
	*w* (µm)	0.56	0.65	0.80	0.72
	$\varepsilon_{V}$ (%)	2.8	4.2	-	-
	*h/d* (nm)	63	91	25	98
400 ms	*A* (µm^2^)	0.071	0.064	0.039	0.099
	*w* (µm)	0.52	0.77	0.50	0.74
	$\varepsilon_{V}$ (%)	4.6	2.8	-	-
	*h/d* (nm)	110	66	49	99

**Table 2 materials-17-00798-t002:** Fitting parameters of each line profile of the As_2_S_3_ film irradiated with cw laser, using a Gaussian function. The volume photoexpansion ($\varepsilon_{V}$) and the height (*h*) of the irradiated lines is also given. The calculated photoexpansion refers to an arsenic trisulfide film with a thickness of 1 µm.

CW Laser Power (mW)		0.5	1	1.5	2
Laser Intensity (kW/cm^2^)		7.1	14.2	21.2	28.3
Laser Scanning Speed	Computed Parameter				
50 µm/s	*A* (µm^2^)	0.003	0.014	0.029	0.021
	*w* (µm)	0.50	0.47	0.41	0.65
	$\varepsilon_{V}$ (%)	0.2	1.0	2.4	1.1
	*h* (nm)	5	23	51	22
100 µm/s	*A* (µm^2^)	0.009	0.023	0.021	0.032
	*w* (µm)	0.49	0.44	0.70	1.24
	$\varepsilon_{V}$ (%)	0.6	1.8	1.0	0.9
	*h* (nm)	15	40	23	22

**Table 3 materials-17-00798-t003:** Fitting parameters of each hillock of the Au\As_2_S_3_ heterostructure irradiated with fs laser pulses, using a Gaussian function. The volume photoexpansion (εV) and height (*h*) of the irradiated microstructures are also given. The calculated photoexpansion refers to an arsenic trisulfide film with a thickness of 1 µm.

Average Laser Power (mW)		0.125	0.25	0.5	1
Laser Fluence (µJ/cm^2^)		22.125	44.25	88.5	176.9
Laser Intensity (MW/cm^2^)		1.475	2.95	5.9	11.8
Exposure Time	Computed Parameter				
100 ms	*A* (µm^2^)	0.009	0.019	0.022	0.011
	*w* (µm)	0.71	0.81	0.94	0.62
	$\varepsilon_{V}$ (%)	0.4	0.8	0.8	0.6
	*h* (nm)	10	18	19	16
400 ms	*A* (µm^2^)	0.010	0.014	0.013	0.013
	*w* (µm)	0.87	0.99	0.89	0.72
	$\varepsilon_{V}$ (%)	0.4	0.5	0.5	0.6
	*h* (nm)	10	12	11	16

**Table 4 materials-17-00798-t004:** Fitting parameters of each line profile of the Au\As_2_S_3_ heterostructure irradiated with cw laser, using a Gaussian function. The volume photoexpansion (εV) and the height (*h*) of the irradiated lines is also given. The calculated photoexpansion refers to an arsenic trisulfide film with a thickness of 1 µm.

CW Laser Power (mW)		0.5	1	1.5
Laser Intensity (kW/cm^2^)		7.1	14.2	21.2
Laser Scanning Speed	Computed Parameter			
50 µm/s	*A* (µm^2^)	0.012	0.013	0.014
	*w* (µm)	1.46	1.34	0.93
	$\varepsilon_{V}$ (%)	0.3	0.3	0.5
	*h* (nm)	7	7	18
100 µm/s	*A* (µm^2^)	0.018	0.023	0.027
	*w* (µm)	0.92	1.20	1.75
	$\varepsilon_{V}$ (%)	0.7	0.6	0.5
	*h* (nm)	16	14	11

## Data Availability

The data presented in this study are available on reasonable request from the corresponding author.

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
