# Peer review of "Fs Laser Patterning of Amorphous As2S3 Thin Films"

_materials, 2024, doi:10.3390/ma17040798_

Round 1

Reviewer 1 Report

Comments and Suggestions for Authors

The manuscript is devoted to a comparison of the processes of transformation of arsenic trisulfide thin films and gold-arsenic trisulfide heterostructures under the influence of UV fs-laser radiation. The results presented in the manuscript complement the data of the same authors presented in their previous publications. In my opinion, after making changes and additions to the text of the manuscript, it can be published in the journal Materials.

There is a serious contradiction in the text of the manuscript. On the one hand, the authors chose the terms photoexpansion, photoevaporation, photodepression, photodissolution, photo-vaporization, photostructural changes, photomodification to describe the effects, and on the other hand, they recognize the decisive role of thermal conductivity in the difference between the processes of swelling and ablation (photoexpansion and photoevaporation) for arsenic trisulfide thin films and gold-arsenic trisulfide heterostructures. If the authors of the manuscript believe that the effects of temperature rise due to the laser heating do not affect the photoexpansion processes in arsenic trisulfide, then this should be confirmed either by additional experiments or by justifying this assumption with the results of other researchers.

It is also necessary to justify the rejection of the term “ablation”, generally accepted for surfaces modified by intense laser radiation, when describing the formation of volcano-like holes on the surface of the film in areas irradiated with high laser powers.

Throughout the text, the values of Laser power (mW) should be replaced by Laser intensity (mW/cm²) or Laser fluence (J/cm2), since in the absence of significant heat diffusion along the surface of the sample, it is Laser intensity/ Laser fluence that determines the geometric parameters of photoexpansion/photoevaporation, whereas the laser power effect is determined by the diameter of the laser beam. The absence of significant thermal diffusion is evidenced by the fact that the dimensions of the hillock and volcano-like hole do not exceed the diameter of the laser beam.

In the tables and text, the number of signs of volume photoexpansion values should be corrected, since the accuracy of determining the film thickness, hillock height, as well as the discrepancy between the actual shape of the hillock and the Gaussian contour does not allow determining the volume photoexpansion value with an accuracy of hundredths of a percent. In addition, when describing the data from the tables, it should be noted that the calculated volume photoexpansion values refer to an arsenic trisulfide film with a thickness of 1 micron.

There are typos in the text of the manuscript - photoevaporization (line 165), increaseded (line 315), width (m) - Figure 6.

Author Response

Dear reviewer,

Thank you for your time and comments. Our answers are in the attached document.

Best regeards,

Alin Velea

Reviewer 2 Report

Comments and Suggestions for Authors

Line 60. “Arsenic trisulphide (As2S3) is intensively studied due to its optical properties and versatility in structural modifications.” Could the authors provide at most recent references to address this statement?

Line 62-63. Authors should check the sentence.

Line 112. “4x106” should be fixed.

Lines 109-116. For clarity laser pulse width should be written.

Figure 1. Proper leveling and scan lines correction of AFM images should be done. It is clearly visible that all 4 images in the right column (for 2 mW) are incorrectly postprocessed.

Line 137. Authors state that “for 1 mW and 400 ms it seems that both effects are present (i.e. the microstructure is half hole, half hillock)”, but did they checked the surface with another/new AFM tip to exclude AFM scanning artefacts?

Line 142.  Authors state that “double structure can be observed” for 1 mW laser power and 100 ms exposure time, but did they checked the sample surface with another/new AFM tip to exclude AFM scanning artefacts induced by the shape of AFM tip?

Figure 2. The depth of the holes increases with the increase of exposure time for 2 mW of laser power, but for the larger exposure times (200 and 400 ms) there is no such correlation. Could the authors mark the cross-sections from figure 2 in the AFM images of figure 1 to exclude the error caused by incorrectly postprocessed AFM images?

Figure 4. The profile is asymmetric for 100 um/s and 2 mW, could the authors show the profiles of scan lines in two opposite directions (trace and retrace) to prove the good tracking of the surface of AFM tip?

Comments on the Quality of English Language

The text is clear, but it would benefit from a thorough review, as there are a few minor mistakes present in the manuscript.

Author Response

(The authors gave the same response as above.)
